# Public Health Concern on Sedentary Behavior and Cardiovascular Disease: A Bibliometric Analysis of Literature from 1990 to 2022

**DOI:** 10.3390/medicina58121764

**Published:** 2022-11-30

**Authors:** Zhen Yang, Sitong Chen, Ran Bao, Ruizhe Li, Kaiming Bao, Renzhi Feng, Ziyi Zhong, Xuebin Wang

**Affiliations:** 1Department of Movement Sciences, KU Leuven, 3001 Leuven, Belgium; 2Institute for Health and Sport, Victoria University, Melbourne, VIC 3011, Australia; 3Priority Research Centre for Physical Activity and Nutrition, School of Education, University of Newcastle, Callaghan, NSW 2308, Australia; 4Department of Rehabilitation Medicine, Nanfang Hospital, Southern Medical University, Guangzhou 510515, China; 5Division of Biokinesiology and Physical Therapy, University of Southern California, Los Angeles, CA 90033, USA; 6School of Sports Medicine and Rehabilitation, Beijing Sport University, Beijing 100084, China; 7School of Biomedical Sciences, University of Leeds, Leeds LS2 9JT, UK; 8Department of Physical Education, Shanghai Jiao Tong University, Shanghai 200240, China

**Keywords:** sedentary behavior, cardiovascular disease, health promotion, bibliometrics, data visualization, mapping knowledge domains, CiteSpace, VOSviewer

## Abstract

*Background and Objectives*: Cardiovascular disease is a long-term threat to global public health security, while sedentary behavior is a modifiable behavior among cardiovascular risk factors. This study aimed to analyze the peer-reviewed literature published globally on sedentary behavior and cardiovascular disease (SB-CVD) and identify the hotspots and frontiers within this research area. *Materials and Methods*: Publications on SB-CVD from 1990 to 2022 were retrieved from the Web of Science Core Collection. CiteSpace and VOSviewer were applied to perform bibliometric and knowledge mapping visualization analyses. *Results*: A total of 2071 publications were retrieved, presenting a gradual growing trend. Authors from the USA topped the list with 748 (36.12%), followed by authors from England (373, 18.01%) and Australia (354, 17.09%). The University of Queensland, Australia, led with 95 (4.5%) publications. The top five active authors were all from Australia, while Dunstan D and Owen N published the most documents (56, 2.7%). A total of 71.27% of the publications received funding, and the United States Department of Health and Human Services provided 363 (17.53%) grants. Public Environmental Occupational Health (498, 24.05%), Sport Sciences (237, 11.44%), and Cardiac Cardiovascular Systems (212, 10.24%) were the three most popular disciplines, while *PLOS One* (96, 4.64%) and *BMC Public Health* (88, 4.25%) were the two most popular journals. Investigations within the SB-CVD research area addressed the entire lifespan, the most popular type of research was the epidemiological study, and the accelerometer was the primary instrument for measuring sedentary behavior. In terms of variables, physical activity and sedentary behavior were the dominant lifestyle behaviors, while obesity and hypertension were common health problems. Occupational physical activity and guidelines are at the frontier and are currently in the burst stage. *Conclusions*: The last three decades have witnessed the rapid development of the SB-CVD research area, and this study provided further research ideas for subsequent investigations.

## 1. Introduction

Cardiovascular disease (CVD), as a non-communicable disease, has been the leading cause of death and disability worldwide since 1990, posing a significant public health issue [1]. CVD is identified as a class of diseases that involve heart or blood vessels, while the common forms of CVD include coronary artery diseases, strokes and transient ischemic attacks, peripheral arterial disease, and aortic disease [2,3]. The World Health Organization (WHO) reported in 2019 that CVD was responsible for 32% of deaths worldwide [4], and was directly responsible for 18.6 million deaths worldwide in 2019 [1]. In terms of the geographical distribution of the burden of CVD, more than three quarters of deaths due to CVD occurred in developing countries [1,4]. Given the accelerating rate of ageing [5], and the difficulty in accessing CVD-related health services in these areas [6], effective and affordable public health actions and interventions against CVD are needed to mitigate the further widening of global health inequalities [7].

Among the various risk factors for CVD, sedentary behavior (SB), which is a modifiable behavioral risk factor, has been of increasing interest to a growing number of stakeholders in recent decades [8]. This complicated human movement behavior is defined as ‘any waking behaviour characterised by an energy expenditure of ≤1.5 metabolic equivalents of task while in a sitting, reclining or lying posture’ [9]. However, in the sport science research area, the term “sedentary” is often applied to describe the incompliant with certain thresholds for moderate to vigorous physical activity (PA) [10]. Meanwhile, SB was also confused with the concept of physical inactivity until the definition of SB in the field of PA for health was standardized [11]. Therefore, in previous studies, the definition of SB varies depending on the time period in which the study took place, the different professions, and other factors.

The sedentary lifestyle is considered a highly prevalent disease in the 21st century due to the serious risk to public health [12]. Prolonged SB is common among children [13], adolescents [14], adults [15], and the elderly [16]. According to the estimation of the WHO, excessive SB is responsible for approximately 3.2 million deaths each year [17]. Considerable evidence demonstrated that high levels of SB increase all-cause and CVD mortality [8,18]. Furthermore, by treating SB as a risk factor independent of PA, previous studies have still reported the association of prolonged SB with high CVD mortality and morbidity [19,20,21]. To address this growing public health issue, the WHO [22], the UK [23], the USA [24], Canada [25], Australia [26], China [27], and other countries have developed a series of evidence-based guidelines on PA and SB. Moreover, several public health intervention trials based on behavior change techniques and wearable devices to reduce SB have been demonstrated to be effective in reducing cardiometabolic risk factors and thus preventing CVD [28,29,30]. Furthermore, PA were often designed as an intervention to replace SB, and previous studies have found an individualized spectrum of PA that affects the body mass index and thus overall probability of metabolic syndrome [31,32].

The majority of scientific research is presented in the form of publications, and bibliometric analysis is a mixed-method analysis of the literature in a given field using mathematical and statistical methods, to assess research trends and present the development frontiers in the given research fields [33]. The purpose of bibliometrics differs from that of systematic reviews and scoping reviews [34]. The primary purpose of a bibliometric analysis in a particular research area is to focus the attention of researchers and funding agencies on under-researched areas, and to help policy makers conduct rational public health policies and decisions [35]. In the public health area, bibliometric analysis has been performed to investigate publications on topics such as vaccine hesitancy [35], global migrant health [34], PA and air pollution [36]. Despite the increasing global burden of SB and CVD on public health systems, and the growing public concern around the world over the past 30 years, hitherto, there has been no bibliometric analysis of global publications on the sedentary behavior and cardiovascular diseases (SB-CVD) research area.

Therefore, this present research aims to provide bibliometric analysis and draw knowledge maps from the global published literature in the SB-CVD research area from 1990 to 2022. Specifically, this present research will identify: (1) the countries, institutions, authors, disciplines, journals and references that contribute significantly to global research on SB-CVD; (2) the current hotspots and primary topics of global research on SB-CVD; and (3) the frontiers in the SB-CVD research area.

## 2. Materials and Methods

### 2.1. Data Source

The Web of Science Core Collection (WoSCC) is a publisher-independent global citation database that provides more than 1.9 billion comprehensive, multidisciplinary, and high-impact cited references [37]. Indeed, other databases exist, but the application of medical or sports science specialized databases such as PubMed and SPORTDiscus may lead to incomprehensive search results, while in the bibliometric analysis, a larger sample size and more comprehensive search results would contribute to greater comprehension and accuracy of the analysis [38]. Furthermore, WoSCC provides older citation information from 1900 onwards, compared to the other interdisciplinary databases like Scopus^@^ [39], and more information about the literature such as countries/regions and affiliations [40]. Moreover, WoSCC, which is identified as the standard database for knowledge mapping [41], has been applied to bibliometric and visual analysis on the topic of Alzheimer’s disease [42], regenerative medicine [43], cardiac rehabilitation [44], PA and aging [45], etc. Additionally, Science Citation Index Expanded, Social Science Citation Index, Art & Humanities Citation Index, and Emerging Sources Citation Index were four indexes in the WoSCC database that have been utilized in previous bibliometric analysis in the public health research area [36]. Accordingly, the above four WoSCC indexes were selected as the data source for this present study.

### 2.2. Search Strategy

To eliminate bias in the search results owing to updates to the WoSCC database, all articles were retrieved on 30 June 2022. The proposed search formula was set to: TS = (“sedentary behaviour” OR “sedentary behavior” OR “sitting” OR “lying” OR “reclining” OR “screen time” OR “TV time” OR “television time” OR “sedentary time” OR “low energy expenditure” OR “seated”) AND TS = (“cardiovascular disease” OR “cardiovascular abnormalities” OR “cardiovascular infections” OR “heart diseases” OR “vascular diseases”), time span = 1990 to 2022. Taking into account previous relevant bibliometric analyses, the inclusion criteria for this study were “article and reviews” [36], and “English literature” [46]. Basic information for each document, including authors, references, countries/regions, institutions, journal sources, keywords, etc., was derived from WoSCC as plain text files. The flow chart of the literature search for this study was presented in Figure 1.

### 2.3. Analysis and Visualization Tools

A total of 2071 articles were part of the bibliometric analysis via CiteSpace 6.1.R2 (64-bit) Basic (Chao-Mei Chen, USA), VOSviewer_1.6.18 (Nees Jan van Eck and Ludo Waltman, Leiden, The Netherlands), and Microsoft Office Excel 2019 (Microsoft Corporation, Washington, USA). CiteSpace, which was developed by Chao-Mei Chen, is a Java-based visualization tool for bibliometrics and data visualization [47]. In this study, CiteSpace was applied to perform literature co-citation analysis, generate knowledge maps of the SB-CVD research area, and find citation bursts of co-cited references and keywords. The co-cited reference is defined as a document being cited by two articles [48], while the analysis of co-citation references is one of the essential metrics of bibliometrics. The visual knowledge map represents the keywords and references with citation bursts, while the occurrence of citation bursts describes the strength of a certain theme frequency. Citation bursts can indicate the intensity and frequency of keywords and references over a period of time to reveal the hotspots and frontiers within a research area [47]. VOSviewer is another visualization tool that was developed by Leiden University [49,50]. Taking into account the advantages of VOSviewer in generating intuitive figures and processing big data [49], it was performed to conduct the distribution and cooperation map analysis of countries/regions, institutions, and authors, as well as the analysis of keyword co-occurrence. The total link strength (TLS) and the sum of times cited (SoTC) were automatically calculated by VOSviewer, while the average citations per item (ACI) were obtained by dividing SoTC by the quantity of published articles. The TLS indicates the number of publications in which two keywords occur together. Excel was utilized to analyze the distribution of annual publications and citations and their changing trends. In previous studies, cubic polynomial functions have been successfully applied to predict publication trends [51,52]. Accordingly, the function model was set as: y = ax^3^ + bx^2^ + cx + d, while y represents the cumulative amounts of publications and x represents the year of publication. Furthermore, Excel was used in this study to analyze the proportion of article types, the distribution of funding agencies, and disciplines based on the data automatically generated from WoSCC.

## 3. Results

### 3.1. Research Type and Annual Distribution Map of the Publications

A total of 2071 publications were included in the final analysis, including 1802 articles and 269 reviews. In the SB-CVD research area, the major publication types were articles, accounting for 87.01%. The annual publications and citations in the SB-CVD research area, which is one of the main indicators to illustrate the development trends in a research area [53], were presented in Figure 2. The years 2000 and 2011 were two milestone years for the SB-CVD research area, with annual publication volumes topping 10 and 50 for the first time, respectively, with publication then peaking in terms of annual volume in the year 2020 (232 publications). Based on data from annual publications 1990–2021, the formula for the trend of annual publication in the SB-CVD research area was obtained by fitting the curve as: y= −0.0135x^3^ + 81.394x^2^ − 163931x + 1E + 08 (R^2^ = 0.8117). The formula predicted that 187 papers would be published in the whole year 2022, while as of 30 June 2022, 74 documents have been published. With the gradual systematization of the SB-CVD research area, and the attention of the general public and academia to the health care field evoked by the COVID-19 pandemic, the SB-CVD research area will present a sustained upward trend.

### 3.2. Distribution of Countries/Regions

As presented in Table 1, the largest number of publications in the SB-CVD research area were from the USA, England, and Australia. Publications from these three countries accounted for more than 70% of the total number of publications, indicating that these three countries had a high interest in the SB-CVD research area. Sweden (87.47) and Belgium (78.70) had the highest ACI, indicating the early development of the SB-CVD research area and the high quality of the research in these two countries. Figure 3 shows the cooperation between countries; the USA frequently cooperated with England, Australia, and Canada; England primarily collaborated with Australia, the USA, and Belgium; and Australia predominantly cooperated with England, the USA, and Belgium.

### 3.3. Distribution of Institutions

The University of Queensland published the largest number of documents in the SB-CVD research area, followed by the University of Sydney and University College London (see Table 2). Only two institutions had an ACI greater than 100; Baker IDI Heart & Diabetes Institute had the highest ACI (175.96), followed by the University of Queensland (126.89). Of the top ten active institutions, six were from Australia, two from the USA, and one each from England and Sweden. Figure 4 presented inter-institutional collaboration in the SB-CVD research area. The University of Queensland frequently cooperated with Baker IDI Heart & Diabetes Institute, Monash University, Deakin University, the University of Sydney, University College London, and Vrije University Amsterdam; and University College London mainly collaborated with the University of Bristol, Harvard Medical School, and the University of Queensland.

### 3.4. Distribution of Authors

Dunstan D and Owen N from Baker IDI Heart & Diabetes Institute contributed to the largest number of publications, followed by Stamatakis E from the University of Sydney (see Table 3). Five of the top ten authors were from Australia, three from England, and one each from the USA and Norway. The collaborative network between the authors was shown in Figure 5, with Dunstan D frequently collaborating with Owen N, Healy G, Winkler E, Salmon J, and Dempsey P; and Stamatakis E mainly cooperated with Hamer M, Bauman A, and Ding D.

### 3.5. Distribution of Funding Agencies

Among all publications in the SB-CVD research area, 1476 studies received at least one grant, accounting for 71.27%. It is clearly presented in Table 4 that the United States Department of Health Human Services and the National Institutes of Health from the USA were the agencies which provided the largest number of grants. Of the top ten funding agencies, five were from the USA, three were from the UK, and one each from Australia and Europe.

### 3.6. Distribution of Disciplines and Journals

#### 3.6.1. Distribution of Disciplines

The documents in the SB-CVD research area were published in a total of 109 WoSCC categories. Public Environmental Occupational Health was the top discipline, accounting for nearly a quarter of the total publications. Sports Sciences, Cardiac Cardiovascular Systems, and Medicine General Internal each accounted for more than 10% of the publications (see Table 5).

#### 3.6.2. Distribution of Journals

The documents retrieved were published in a total of 673 journals. As shown in Table 6, the most popular journal in the SB-CVD research area was *PLOS One*, followed by *BMC Public Health*, and the *International Journal of Behavioral Nutrition and Physical Activity*. The top ten journals published nearly 25% of the articles in the SB-CVD research area. The *American Journal of Preventive Medicine* has the highest ACI, followed by the *British Journal of Sports Medicine*. Combining ACI with impact factor and journal citation reports, the *American Journal of Preventive Medicine* and the *British Journal of Sports Medicine* were presumably high-quality journals in the SB-CVD research area. Furthermore, in the top ten journals in the SB-CVD research area, five were from the USA, four from England, and one from Switzerland.

### 3.7. Highly Cited Publications and Co-Cited References

#### 3.7.1. Highly Cited Literatures

The top ten most highly cited publications in the SB-CVD research area were clearly presented in Table 7. Among the top ten most highly cited publications, 7 were articles, while 3 publications were reviews. Dahlöf, B et al. published an article in *The Lancet* in 2002, and received the most citations [54]. This randomized trial compared the differences between Losartan and atenolol in preventing cardiovascular morbidity and death, while sitting blood pressure was identified as a measure of relevant outcomes [54]. The article ‘Physical activity and public health in older adults: recommendation from the American College of Sports Medicine and the American Heart Association’, which was published in *Medicine & Science in Sports & Exercise* in 2007, obtained the second highest number of citations [55]. Nelson, M. E et al. recommended that older adults perform moderate intensity PA and muscle-strengthening exercise, as well as reduce SB to promote mental, physical, and cardiovascular health after reviewing existing consensus statements and primary evidence [55]. The third most cited publication was an article that was published in *The New England Journal of Medicine* in 2008 [56]. Beckett NS et al. conducted a multicenter placebo-controlled trial and reported that indapamide (extended release), with or without perindopril, could effectively reduce sitting blood pressure in people aged 80 years or older [56].

#### 3.7.2. Co-Cited Reference

The analysis of co-citation reference can be performed by analyzing the number of publications and citations and their changes over time to generate clusters, thus exploring the hotspots and trends in the SB-CVD research area. This present study constructed and presented clusters of the SB-CVD research area using mapped knowledge domains and timeline views in CiteSpace. The Log-Likelihood Ratio algorithm was applied in this study to create the label clusters, taking into account previous research and the merits of the Log-Likelihood Ratio algorithm that this strategy could cover the “uniqueness and coverage” of all labels [36]. The modularity value (Q) and the weighted mean silhouette value (S) were applied to assess the validity of the clusters, while Q > 0.5 with S > 0.7 is the well-accepted validity threshold for a cluster. In this clustering, Q = 0.7079 with S = 0.8634 validated the effectiveness and reasonableness of the present clustering strategy. The cluster view of the knowledge map based on analysis of the co-cited reference in the SB-CVD research area from 1990 to 2022 was presented in Figure 6. The largest of the nine clusters was SB (#0), followed by cluster #1 (work) and cluster #2 (obesity). The timeline view of the development and evolution of each cluster was shown in Figure 7.

The top twenty co-cited references in the SB-CVD research area were presented in Figure 8, with the most co-cited references having the strongest citation bursts. Burst co-citation of the reference was shown in red on the blue timeline. The article: “Sitting time and mortality from all causes, cardiovascular disease, and cancer”, which was published by Katzmarzyk in 2009, received the strongest citation bursts (56.51) [57].

### 3.8. Research Hotspots and Frontier Analysis

#### 3.8.1. Research Hotspots

Keywords, as an imperative ingredient of a scientific article, can be several words that highly condense the core findings of the full text. The number of occurrences of keywords and their change over time indicate hotspots and development directions in a particular research area. This present study conducted a keyword co-occurrence analysis using VOSviewer to identify hotspots and development directions in the SB-CVD research area. After removing duplications caused by the difference in spelling between British English and American English, Table 8 was obtained. In Table 8, PA, SB, exercise, and sleep were frequent keywords as modifiable behaviors. In terms of diseases and health problems, CVD, obesity, hypertension, and metabolic syndrome were the most common keywords that emerged. Regarding types of study, epidemiology was the most frequent keyword, while accelerometer was the most common keyword as a measurement instrument. Blood pressure, mortality, and sitting time were the most frequent outcome indicators.

#### 3.8.2. Research Frontiers

The top twenty keywords with the strongest citation bursts in the SB-CVD research area from 1990 to 2022, which were based on the keyword co-occurrence network, were shown in Figure 9. Burst co-occurrence of the keyword was shown in red on the blue timeline. Blood pressure was the keyword with the longest burst duration and the strongest citation burst, followed by obesity, television viewing time, lifestyle, type 2 diabetes mellitus, and coronary heart disease. Viewing time, meta-analysis, and people were recently occurring keywords, albeit for a short burst duration. Guideline is the current frontier in the SB-CVD research area, which has recently been within the burst stage.

## 4. Discussion

### 4.1. Main Findings

In summary, SB-CVD research experienced an initial phase with low publication volume from 1990 to 2000, a development phase with fluctuating increase in publication volume from 2001 to 2011, and a hot phase from 2012 to 2022. The trend in publication volume within the SB-CVD research area is similar to other related studies [73,74], probably due to the increased focus on the burden of SB and CVD in medical and scientific communities [75,76]. The research in the SB-CVD area started in the western developed countries, and China and Brazil were the only developing countries among the top ten most active countries. This indicates that research in the SB-CVD research area started earlier and is more mature in western developed countries, while developing countries are also beginning to focus on this area, which is consistent with the results of previous related studies [52,77]. The institutions with high volumes of publication were mainly from developed western countries, and collaboration in the SB-CVD research area was mainly inter-country. This is a common situation, as researchers from the USA, Europe, and Australia are typically ranked highly in most scientific disciplines [66,78,79,80]. Regrettably, no active countries or institutions from Africa and the Middle East were observed, despite more than three quarters of deaths due to CVDs occurring in developing countries [1,4]. This may be due to the fact that in SB-CVD studies, SB is mainly measured by accelerometers or inclinometers [70,76,81], and these expensive instruments represent a significant challenge for research investment in developing countries. Moreover, the cost-effective self-reported SB scales lack local adaptation to the language and culture of some developing countries [82]. In terms of disciplines, SB-CVD research was interdisciplinary and includes public health, sports science, and cardiovascular systems, which was also reflected in the most popular journals within this area. The journal that published the largest number of documents in the SB-CVD research area, *PLOS One*, is a comprehensive journal. However, there are four journals belonging to public health or preventive medicine among the top ten most popular journals. Furthermore, the top ten most popular journals within this research area all belonged to the Q1 or Q2, indicating the high quality of publication within this research area.

### 4.2. Research Hotspots

To guide research on PA, Welk adapted the behavioral epidemiology framework for PA that demonstrates how PA can be understood and how different types of research can be used to facilitate it [83]. Subsequently, this framework was applied to SB. Combining this framework with the analysis of co-cited reference, and the analysis of keyword co-occurrence, the hotspots of populations, measurement instruments, modifiable behaviors, and health issues in the SB-CVD research area were identified.

It is shown in Figure 10 that the framework centers on measuring PA/SB, while accurate estimation of potential PA/SB is a prerequisite for follow-up studies. There are various instruments with different levels of validity and acceptability that can be applied to measure PA/SB, ranging from very precise but intrusive measures such as doubly labelled water, to more user-friendly device-based measures, such as accelerometers, to self-reported measures such as questionnaires and diaries [83]. When selecting an appropriate instrument, it is important to not only consider its limitations and advantages, but also its suitability for practical purposes [84]. In the SB-CVD research area, the accelerometer is the measurement instrument hotspot to assess the duration of SB, and to explore the relationship between SB and CVD, or risk factors in cross-sectional or longitudinal studies [85,86]. Instruments for the subjective measurement of SB, such as self-report questionnaires, may not be widely adopted due to their lack of reliability and validity. However, given their economic advantages and the high prevalence of CVD in developing countries, subjective measures could be applied to local epidemiological studies to reduce global health inequalities.

The research type hotspot in the SB-CVD research area was the epidemiological study. A 4.9-year follow-up study reported a dose–response relationship between longer total sedentary time and longer sedentary bouts, and an increased risk of CVD in older women [76]. Some research on the elaboration of mechanisms, as well as intervention studies, also exist. Chauntry et al. identify excessive stress response as a possible mechanism linking SB to CVD [87], while wearable devices were proposed to be effective in reducing SB and decreasing blood pressure in older adults [30]. In the SB-CVD research area, in addition to SB, PA is another behavior hotspot that cannot be overlooked. The three movement continuums of SB, PA, and sleep make up a person’s 24-h movement behavior [88], while previous research in the SB-CVD research area found that SB is independent of PA and a separate risk factor for a range of health outcomes [71]. Recent studies have used the novel isotemporal substitution paradigm and found that reassigning SB to moderate to vigorous PA was associated with a reduction in cardiovascular risk factors [89]. CVD covers quite a few diseases or health problems, while the most frequently mentioned in the SB-CVD research area were obesity and hypertension. Previous research proposed that prolonged SB was associated with the risk of obesity in children [90], adolescents [91], adults [92], and elderly [93]. Hypertension has also emerged with great frequency in the SB-CVD research area [76,94], probably owing to its convenience of measurement and its predictive value as a risk factor for CVD [95]. Furthermore, the population hotspots in the SB-CVD research area cover practically the entire life, including children [96], adolescents [97], the working population [95], and the elderly [98].

Despite the fact that studies in the SB-CVD research area cover the population across the lifespan, as well as several health outcomes, there exists limitation in the research types. Epidemiological studies have indisputably contributed the majority of publications in the SB-CVD research area. Regrettably, there is a lack of basic and intervention research within this field, which has also been reported by previous related research [36,44]. Although there exist several mechanisms interpretational studies [87], they primarily investigated the potential relationship based on epidemiological rather than experimental evidence. Current basic research has only explored the underlying mechanisms by which SB influences vascular function through hemodynamic stimuli [99], inflammation and reactive oxygen species [100], and metabolic markers [101].

Nevertheless, with the growing body of epidemiological and interventional evidence, basic research in the SB-CVD research area is urgently needs more attention from researchers.

### 4.3. Research Frontiers

Analysis of the co-cited reference reveals that cluster #5, occupational PA, was the frontier in the SB-CVD research area. With the progress of industrialization and advances in computer science, more and more people are moving from physical to mental work, leading to an increase in SB in the workplace. The office may be a distinctive scenario where the relationship between PA/SB and CVD is the exact opposite of the other scenarios. Previous studies have shown that more SB was associated with better cardiorespiratory health [102]. Quinn et al. proposed that people who were physically active in the office were at a higher risk of CVD than those who never participate in occupational PA [103]. In other scenarios, more PA and less SB were associated with lower CVD prevalence and mortality [104,105]. The causes and mechanisms of this public health paradox remain underexplored. Furthermore, workplaces appear to be well suited for multicenter randomized trials. Several public health interventions, which aimed to reduce the risk of CVD through occupational PA intervention toolkits, have been reported [106,107].

The guideline is the frontier of the SB-CVD research area as identified by the analysis of keyword co-occurrence. The WHO [22], the UK [23], the USA [24], Canada [25], Australia [26], China [27], and other countries have developed or updated a series of evidence-based guidelines on PA and SB. WHO 2020 guidelines on PA and SB recommend a limitation of SB across the lifespan, based on evidence of the relationship between SB and CVD in different age groups (children, adolescents, adults, and older adults) [22]. Moreover, modified 24-hour home-based movement behaviors guidelines based on CVD patient characteristics have been developed [100]. In addition to public education and clinical practice through the guidelines, some epidemiological studies identified compliance with the guidelines as an outcome variable [108,109]. Notably, in contrast to explicit quantitative targets for PA, most SB guidelines did not quantify thresholds for SB due to a lack of developed evidence, but rather replace them with generic non-quantitative reductions [110]. This indicates that in the SB-CVD research area, it is advisable to develop evidence-based SB guidelines for patients with different CVDs, and to provide explicit quantitative targets for SB guidelines through high-quality epidemiological studies.

### 4.4. Limitations and Further Recommendations

We acknowledge that there are limitations to this present study. First, only four indexes from WoSCC were identified as data sources, whereas no other databases (e.g., PubMed, Scopus) were included, which may lead to not covering all publications within the SB-CVD research area. However, WoSCC, as one of the most comprehensive high-quality databases in the world, contains sufficient publications for bibliometric analysis. Moreover, the existing bibliometric and visual knowledge mapping tools do not facilitate cross-database analysis. In the future, researchers and software developers will be expected to upgrade relevant tools and algorithms to realize the de-duplication and merging of data from different databases for analysis. The second limitation is the linguistic bias resulting from the fact that only documents published in English were included in this study. Third, the WoSCC does not distinguish between different spellings or abbreviations of authors, institutions, or keywords, which may lead to some duplication or omission. However, there is no perfect search and research strategy that can completely avoid false positive or false negative results for a research question. Finally, due to the limitations of the bibliometric research methodology itself, which focuses on completeness of literature, there may have been publications in this study that were only related to the words “sitting” rather than the content itself. Further refinement of the search strategy may be required in subsequent studies.

## 5. Conclusions

This study illustrated the research progress, hotspots, and research frontiers of the SB-CVD research area from 1990 to 2022 through bibliometrics and visual knowledge mapping techniques. After three decades of development, SB-CVD research has reached a stable stage with a steady increase in the number of citations. Within the SB-CVD research area, institutions and researchers from western developed countries contributed the majority of publications. SB-CVD is a multidisciplinary research area and important disciplines, journals, funding agencies, and representative references within this area were identified. Investigations within the SB-CVD research area addressed the entire lifespan, the most popular type of research was the epidemiological study, and the accelerometer was the primary instrument used for measuring SB. In terms of the variables, PA and SB were the dominant lifestyle behaviors, while obesity and hypertension were the most common CVD-related health problems. Occupational PA and guidelines are at the frontiers of the SB-CVD research area and are currently in the burst stage. The above findings provided further research ideas for subsequent investigation on SB-CVD and provided a reference for international collaboration.

## Figures and Tables

**Figure 1 medicina-58-01764-f001:**
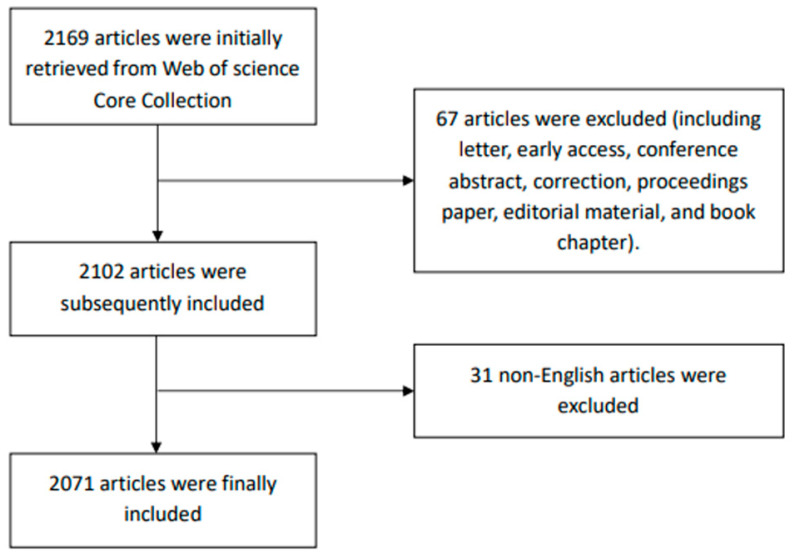
Flowchart of literature research.

**Figure 2 medicina-58-01764-f002:**
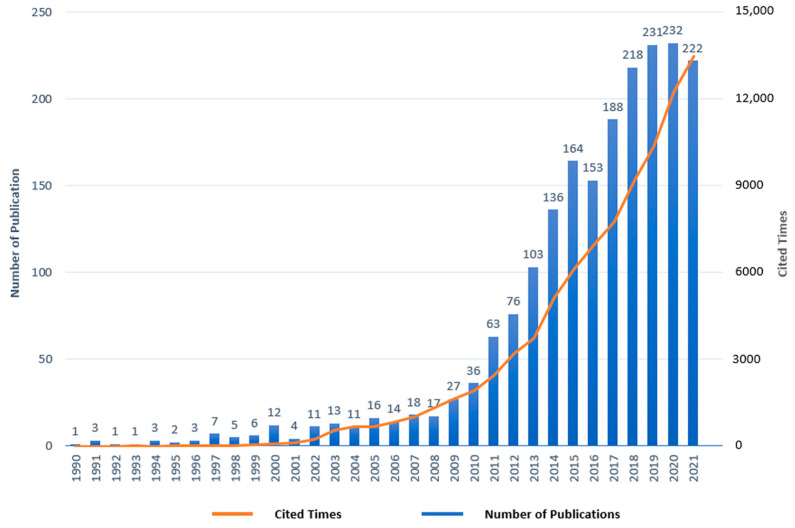
Yearly changes of publications and citations of sedentary behavior and cardiovascular diseases (SB-CVD) publications.

**Figure 3 medicina-58-01764-f003:**
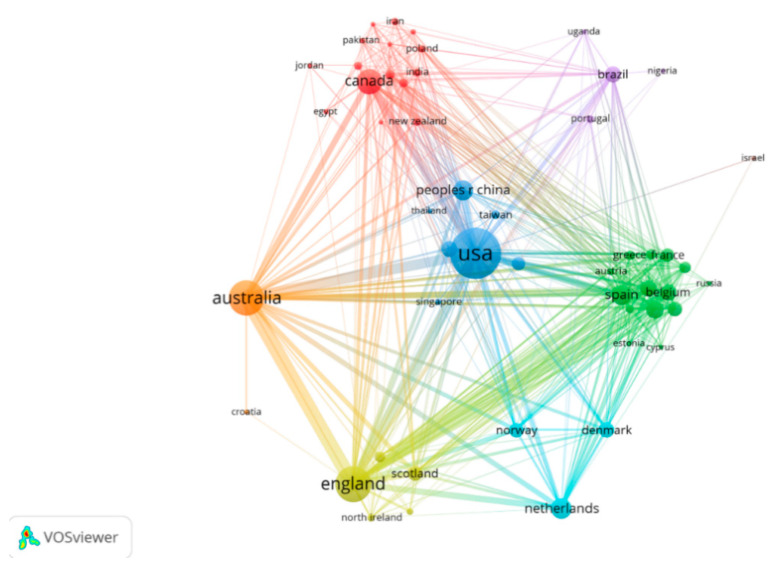
Inter-country cooperation map of SB-CVD research area from 1990 to 2022.

**Figure 4 medicina-58-01764-f004:**
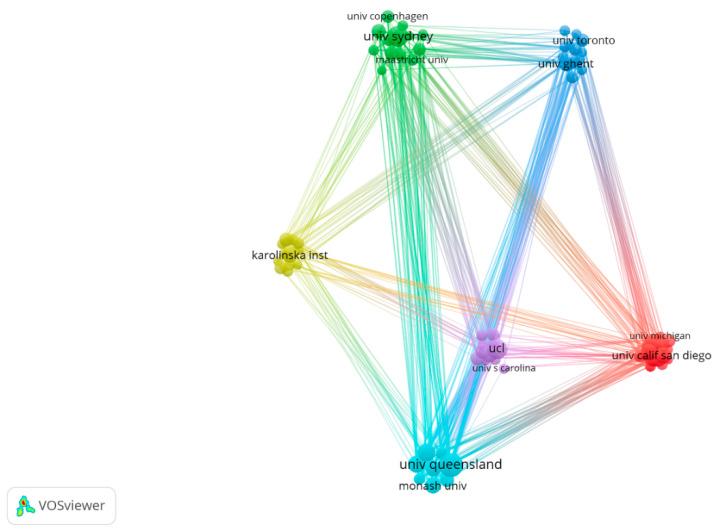
Inter-institutional collaboration map of SB-CVD research area from 1990 to 2022.

**Figure 5 medicina-58-01764-f005:**
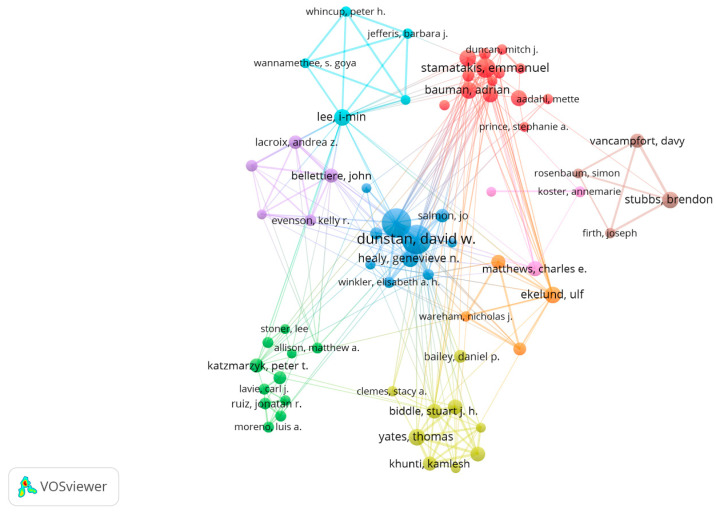
Collaboration map of authors in the SB-CVD research area from 1990 to 2022.

**Figure 6 medicina-58-01764-f006:**
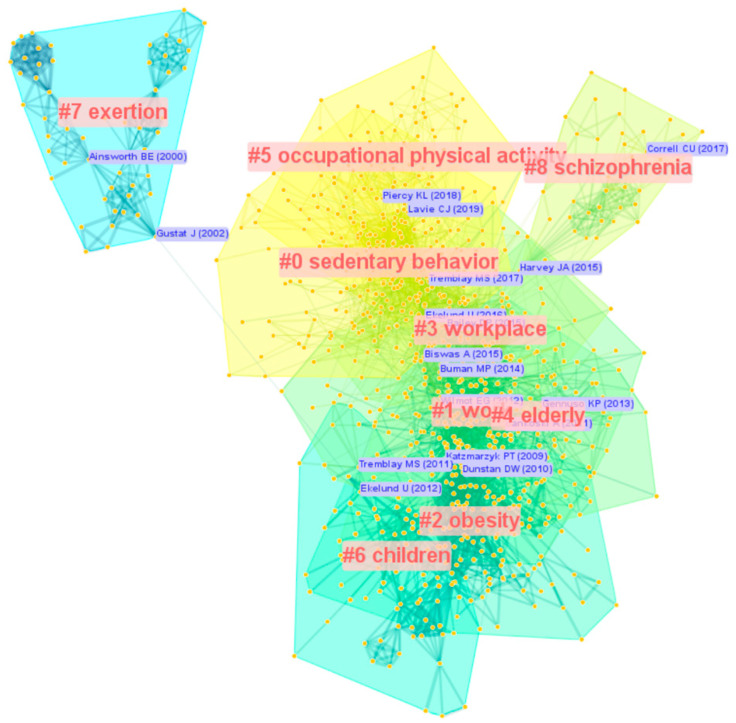
The cluster view of the knowledge map based on analysis of co-cited reference in the SB-CVD research area from 1990 to 2022.

**Figure 7 medicina-58-01764-f007:**
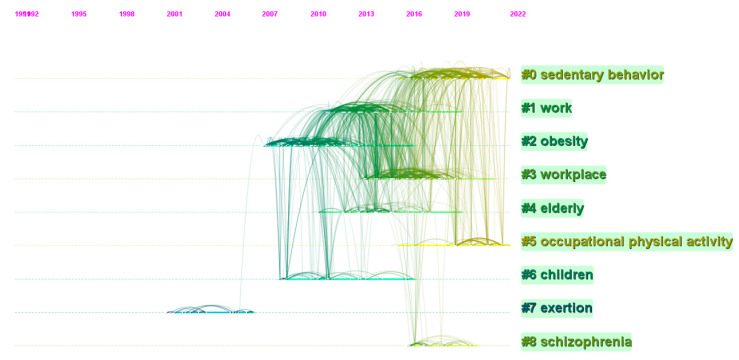
The timeline view of the knowledge map based on analysis of co-cited reference in the SB-CVD research area from 1990 to 2022.

**Figure 8 medicina-58-01764-f008:**
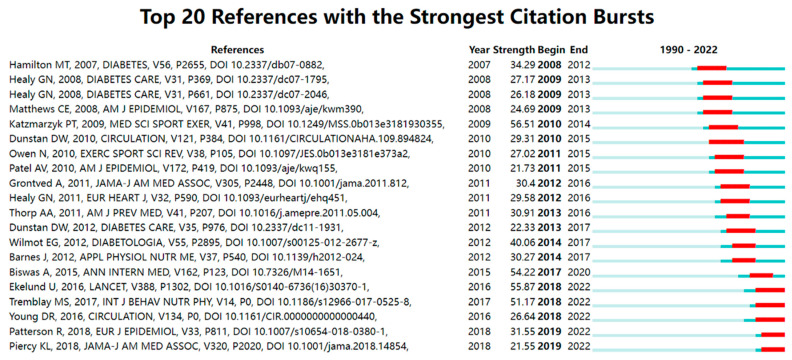
Top twenty references with the strongest citation bursts in the SB-CVD research area from 1990 to 2022 [8,10,18,20,56,58,59,60,61,62,63,64,65,66,67,68,69,70,71,72].

**Figure 9 medicina-58-01764-f009:**
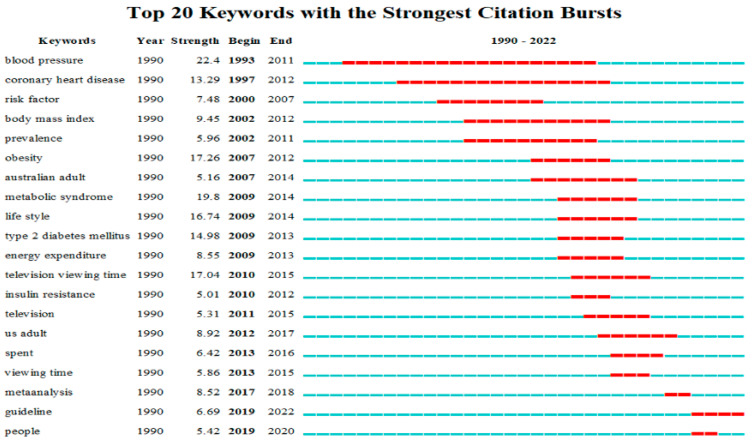
Top twenty keywords with the strongest citation bursts in the SB-CVD research area from 1990 to 2022.

**Figure 10 medicina-58-01764-f010:**
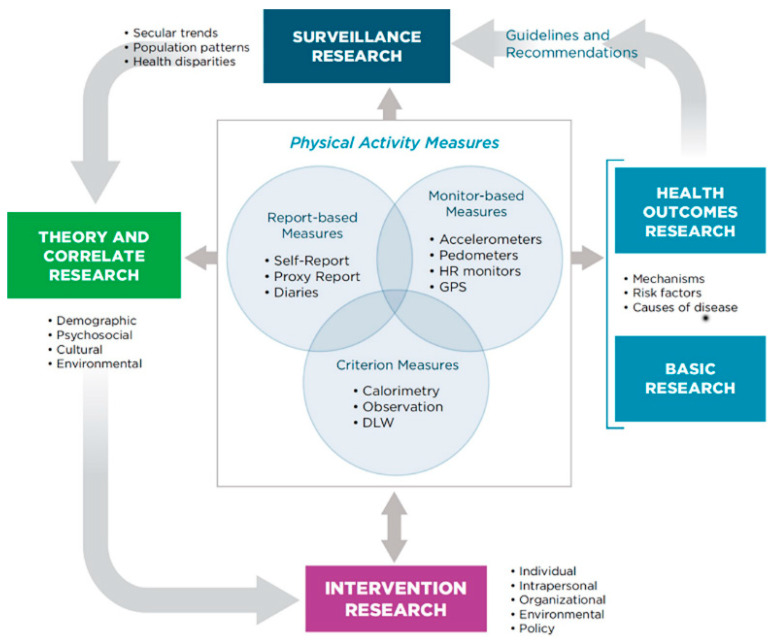
Behavioral epidemiology framework adapted for PA and SB research.

**Table 1 medicina-58-01764-t001:** Top ten active countries/regions in the SB-CVD research area from 1990 to 2022.

Rank	Country	Quantity	Percentage (%)	ACI	TLS
1	USA	748	36.12%	60.52	507
2	England	373	18.01%	65.57	530
3	Australia	354	17.09%	66.60	462
4	Canada	187	9.03%	57.58	185
5	Netherlands	125	6.04%	40.54	183
6	Peoples R China	118	5.70%	41.49	119
7	Spain	115	5.55%	30.00	218
8	Sweden	86	4.15%	87.47	182
9	Brazil	77	3.72%	40.62	119
=10	Belgium	76	3.67%	78.70	231
=10	Germany	76	3.67%	38.25	160
=10	Japan	76	3.67%	20.97	72

ACI: average citations per item; TLS: total link strength; %: percentage of the quantity of work per country out of the 2071 articles included in the literature research; =: same quantity of articles.

**Table 2 medicina-58-01764-t002:** Top ten active institutions in the SB-CVD research area from 1990 to 2022.

Rank	Institution	Country	Quantity	Percentage (%)	ACI	TLS
1	The University of Queensland	Australia	95	4.59%	126.89	345
2	The University of Sydney	Australia	73	3.52%	71.79	202
3	University College London	England	65	3.14%	62.71	141
4	Deakin University	Australia	53	2.56%	79.55	202
5	Baker IDI Heart & Diabetes Institute	Australia	48	2.31%	176.96	198
6	University of California San Diego	USA	45	2.17%	26.51	165
=7	University of Illinois	USA	44	2.12%	57.34	148
=7	Australian Catholic University	Australia	43	2.08%	23.63	210
9	Monash University	Australia	42	2.03%	90.07	207
10	Karolinska Institute	Sweden	42	2.03%	51.29	117

ACI: Average citations per item; TLS: Total link strength; %: percentage of the quantity of work per institution out of the 2071 articles included in the literature research; =: same quantity of articles.

**Table 3 medicina-58-01764-t003:** Top 10 prolific authors in the SB-CVD research area from 1990 to 2022.

Rank	Author	Country	Institution	Quantity	Percentage (%)	ACI	TLS
=1	Dunstan D	Australia	Baker IDI Heart & Diabetes Institute	56	2.70%	129.68	123
=1	Owen N	Australia	Baker IDI Heart & Diabetes Institute	56	2.70%	172.95	113
3	Stamatakis E	Australia	The University of Sydney	29	1.40%	82.03	63
=4	Bauman A	Australia	The University of Sydney	23	1.11%	147.26	37
=4	Healy G	Australia	The University of Queensland	23	1.11%	231.09	68
=4	Stubbs B	England	King’s College London	23	1.11%	62.96	34
=4	Ekelund U	Norway	Norwegian School of Sport Sciences	23	1.11%	142.43	45
=4	Hamer M	England	Loughborough University	23	1.11%	81.39	35
=9	Lee I	USA	Harvard Medical School	22	1.06%	110.64	67
=9	Yates T	England	University of Leicester	22	1.06%	50.27	72

ACI: Average citations per item; TLS: Total link strength; %: percentage of the quantity of work per author out of the 2071 articles included in the literature research; =: same quantity of articles.

**Table 4 medicina-58-01764-t004:** The top ten funding agencies in the SB-CVD research area from 1990 to 2022.

Rank	Agency	Quantity	Percentage (%)	Country/Region
1	United States Department of Health and Human Services	363	17.53%	USA
2	National Institutes of Health	357	17.24%	USA
3	European Commission	188	9.08%	Europe
4	National Heart Lung Blood Institute	170	8.21%	USA
5	UK Research Innovation	143	6.91%	UK
6	UK Medical Research Council	133	6.42%	UK
7	National Health and Medical Research Council of Australia	114	5.51%	Australia
=8	National Institute for Health Research	82	3.96%	UK
=8	National Cancer Institute	82	3.96%	USA
10	National Institute on Aging	76	3.67%	USA

%: percentage of the quantity of funded work per agency out of the 2071 articles included in the literature research; =: same quantity of articles.

**Table 5 medicina-58-01764-t005:** Top ten Web of Science Core Collection categories in the SB-CVD research area from 1990 to 2022.

Rank	WoSCC Categories	Quantity	Percentage (%)
1	Public Environmental Occupational Health	498	24.05
2	Sport Sciences	237	11.44
3	Cardiac Cardiovascular Systems	212	10.24
4	Medicine General Internal	209	10.09
5	Nutrition Dietetics	171	8.26
6	Peripheral Vascular Disease	160	7.73
7	Physiology	134	6.47
8	Endocrinology Metabolism	123	5.94
9	Multidisciplinary Sciences	116	5.60
10	Environmental Sciences	70	3.38

%: percentage of the quantity of work per WoSCC category out of the 2071 articles included in the literature research.

**Table 6 medicina-58-01764-t006:** The top ten most popular journals in the SB-CVD research area from 1990 to 2022.

Rank	Journal	Quantity	Percentage (%)	ACI	IF (2021)	Country	JCR
1	PLOS One	96	4.64	49.26	3.752	United States	Q2
2	BMC Public Health	88	4.25	29.95	4.135	England	Q2
3	International Journal of Behavioral Nutrition and Physical Activity	62	2.99	52.87	8.915	England	Q1
4	International Journal of Environmental Research and Public Health	61	2.95	6.64	4.614	Switzerland	Q2/Q1 *
5	Medicine & Science in Sports & Exercise	48	2.32	73.81	6.289	United States	Q1
6	Preventive Medicine	39	1.88	27.51	4.637	United States	Q2
7	Journal of Physical Activity & Health	36	1.74	13.81	3.000	United States	Q2 *
8	BMJ Open	33	1.59	10.36	3.006	England	Q2
9	American Journal of Preventive Medicine	26	1.26	117.88	6.604	United States	Q1/Q1 *
10	British Journal of Sports Medicine	25	1.21	94.28	18.473	England	Q1

ACI: Average citations per item; IF: Impact Factor; JCR: Journal Citation Reports; * Represents SSCI, and the default for the JCR is SCI; %: percentage of the quantity of work per journal out of the 2071 articles included in the literature research.

**Table 7 medicina-58-01764-t007:** Top ten most highly cited publications in the SB-CVD research area from 1990 to 2022.

Rank	Title	Journal	Type	Author	Year	SoTC
1	Cardiovascular morbidity and mortality in the losartan intervention for endpoint reduction in hypertension study (LIFE): a randomised trial against atenolol	The Lancet	Article	Dahlof B. et al.	2002	3829
2	Physical activity and public health in older adults: recommendation from the American College of Sports Medicine and the American Heart Association	Medicine & Science in Sports & Exercise	Article	Nelson M.E. et al.	2007	2659
3	Treatment of hypertension in patients 80 years of age or older	The New England Journal of Medicine	Article	Beckett N.S. et al.	2008	1972
4	Sedentary time and its association with risk for disease incidence, mortality, and hospitalization in adults: a systematic review and meta-analysis	Annals of Internal Medicine	Review	Biswas A. et al.	2015	1422
5	Too much sitting: the population health science of sedentary behavior	Exercise and Sport Sciences Reviews	Review	Owen N. et al.	2010	1382
6	Does physical activity attenuate, or even eliminate, the detrimental association of sitting time with mortality? A harmonised meta-analysis of data from more than 1 million men and women	The Lancet	Article	Ekelund U. et al.	2016	1190
7	Television watching and other sedentary behaviors in relation to risk of obesity and type 2 diabetes mellitus in women	JAMA-Journal of The American Medical Association	Article	Hu F.B. et al.	2003	1148
8	Systematic review of sedentary behaviour and health indicators in school-aged children and youth	International Journal of Behavioral Nutrition and Physical Activity	Review	Tremblay M.S. et al.	2011	1144
9	Role of low energy expenditure and sitting in obesity, metabolic syndrome, type 2 diabetes, and cardiovascular disease	Diabetes	Article	Hamilton M.T. et al.	2007	1059
10	Sitting time and mortality from all causes, cardiovascular disease, and cancer	Medicine & Science in Sports & Exercise	Article	Katzmarzyk P.T. et al.	2009	1054

SoTC: sum of times cited.

**Table 8 medicina-58-01764-t008:** Top twenty keywords in the SB-CVD research area from 1990 to 2022.

Rank	Keyword	Occurrence	TLS	Rank	Keyword	Occurrence	TLS
1	Physical activity	530	1612	11	Sedentary lifestyle	72	193
2	Sedentary behaviour	430	1218	12	Mortality	69	257
3	Cardiovascular disease	220	674	13	Sedentary	61	190
4	Exercise	190	603	14	Metabolic syndrome	56	172
5	Obesity	134	400	15	Sitting time	53	156
6	Accelerometer	132	398	16	Diabetes	50	186
7	Hypertension	114	257	17	Sedentary time	50	137
8	Epidemiology	91	266	18	Risk factors	48	146
9	Blood pressure	78	175	19	Sleep	40	168
10	Sitting	74	238	20	Prevention	40	138

TLS: Total link strength.

## Data Availability

Not applicable.

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
