# Peer review of "Public Health Concern on Sedentary Behavior and Cardiovascular Disease: A Bibliometric Analysis of Literature from 1990 to 2022"

_medicina, 2022, doi:10.3390/medicina58121764_

Round 1
Reviewer 1 Report
In the article “Public Health Concern on Sedentary Behavior and Cardiovascular Disease: A Bibliometric Analysis of Literature from 1990 to 2022”, Zhen Yang and group sought to evaluate the existing body of work on sedentary behavior and cardiovascular disease (SB-CVD). The authors highlight the burden of sedentary behavior on public health, the countries and institutes leading research in SB-CVD, the scope and frontiers of the field, and the funding agencies subsidizing the works. Overall, the authors give a comprehensive, complete and well-written bibliometric work on SB-CVD research, especially as it pertains to behavioral studies.
Major concern:
1. Please discuss the lack of basic research in the SB-CVD arena.
Minor concerns:
1. Please clearly define SB-CVD in the abstract.
2. A few grammatical issues were noted throughout the text; please address them.
3. As mentioned in the Methods section, Total link strength (TLS) was calculated automatically by VOSviewer. Can you please give a brief interpretation on its importance in your work?
4. Percentage (%) is a common variable in Tables 1-6. However, these are capturing different information per table. Can you please specify the information per each? For example, the variable in Table 1 is the percentage of the quantity of work per country out of the 2071 articles included in the literature research.
5. In Table 7, how does the work by Dahlof et al pertain to sedentary behavior? Please clarify.
6. I find Figure 8 redundant as it mentions ‘sedentary behavior’ and ‘cardiovascular disease’ as keywords, when they are given in SB-CVD. Please clarify how this adds value to your work.
Author Response
To Reviewer 1:
Major concerns:
Q1: Please discuss the lack of basic research in the SB-CVD arena.
A1: Thank you for your practical suggestions, we acknowledge that I did not have enough discussion on the basic research in the SB-CVD arena.
Location:
Minor concerns:
Q2: Please clearly define SB-CVD in the abstract.
A2: Thank you for your practical suggestions, we neglected to mention the full spelling when abbreviations are first mentioned in the abstract, which we have modified in the manuscript.
Location: Line 21-22.
Q3: A few grammatical issues were noted throughout the text; please address them.
A3: Thank you for your practical suggestions, we have double-checked our manuscript and revised the grammar with the help of an English native speaker.
Location: Line 47, 51, 52, 59, 70, 80, 89, 103-104 et al.
Q4: As mentioned in the Methods section, Total link strength (TLS) was calculated automatically by VOSviewer. Can you please give a brief interpretation on its importance in your work?
A4: Thank you for your practical suggestions. Total link strength is definitely an important metric in bibliometric analysis, and I have added a brief interpretation in the methods section.
Location: Line 159-160.
Q5: Percentage (%) is a common variable in Tables 1-6. However, these are capturing different information per table. Can you please specify the information per each? For example, the variable in Table 1 is the percentage of the quantity of work per country out of the 2071 articles included in the literature research.
A5: Thank you for your practical suggestions, we have modified them in our manuscript which will make the results more clearly.
Location: 198-199, 215-216, 228-229, 240-241, 250-251, 265-266.
Q5: In Table 7, how does the work by Dahlof et al pertain to sedentary behavior? Please clarify.
A5: This is a good question. We reviewed Dehlof's full paper and found that its relationship with cardiovascular disease is clear. However, in terms of sedentary behaviour, it only relates to one possible word in 'sitting blood pressure'. This may be due to the inclusion of "sitting" in our search terms, which is a limitation of the bibliometric study methodology itself. We have added this consideration to the limitations.
Location: Line 482-486.
Q6: I find Figure 8 redundant as it mentions ‘sedentary behavior’ and ‘cardiovascular disease’ as keywords, when they are given in SB-CVD. Please clarify how this adds value to your work.
A6: Thank you for your practical suggestions. We double-checked the content of Figure 8 and Table 8, and we found that Figure 8 was redundant. Therefore, we deleted Figure 8.
Reviewer 2 Report
1. Author should include specific information about sedentary behavior. What defines SB as continuous work hours, high dietary intake and poor mobilisation, long sleep hours and low-energy activities. Metabolic equivalents are strictly defined in terms of energy expenditure by physical activities. On the other side, sedentary behaviour is equated with physical inactivity which is not applicable to large section of office professionals.
2. Over a period of time, it is observed that there is an individualised spectrum of physical activity that affects the BMI and thus overall probability of metabolic syndrome.
Author Response
To Reviewer 2:
Q1: Author should include specific information about sedentary behavior. What defines SB as continuous work hours, high dietary intake and poor mobilisation, long sleep hours and low-energy activities. Metabolic equivalents are strictly defined in terms of energy expenditure by physical activities. On the other side, sedentary behaviour is equated with physical inactivity which is not applicable to large section of office professionals.
A1: Thank you for your practical suggestions. I understand that the term “sedentary” currently has two separate and contradictory operational definitions. In the physical activity and health research area, where I obtained education and scientific research training, sedentary behaviours are typically defined by both low energy expenditure (e.g., resting metabolic rate, typically ≤1.5 metabolic equivalents (METs)) and a sitting or reclining posture (Owen et al. 2010; Pate et al. 2008; Tremblay et al. 2010). In this context, a person may be described as sedentary if they engage in a large amount of sedentary behaviour.
In contrast, in the sport and exercise literature the term sedentary is frequently used to describe the absence of some threshold of MVPA (Church et al. 2009; Melanson et al. 2009; Mullen et al. 2011; Sims et al. 2012; Smith et al. 2010). Thus, it is common for researchers in this field to describe a participant as sedentary because they are not meeting physical activity guidelines. Hence, many exercise studies include a “sedentary control group” or refer to their participants as coming from a “sedentary population” because of their lack of physical activity without actually measuring or assessing their level of sedentary behaviour.
Considering that this study addresses sedentary behaviour and cardiovascular disease, a public health issue that requires multidisciplinary collaboration, I believe that definitions from the fields of exercise for health and exercise physiology need to be combined in order to extend the results of this study to a wider range of stakeholders and clinical populations.
Here is the reference which could support my answers:
Sedentary Behaviour Research Network. Letter to the Editor: Standardized use of the terms “sedentary” and “sedentary behaviours”. Applied Physiology, Nutrition, and Metabolism. 37(3): 540-542. https://doi.org/10.1139/h2012-024
Owen N., Healy G.N., Matthews C.E., and Dunstan D.W. 2010. Too much sitting: the population health science of sedentary behavior. Exerc. Sport Sci. Rev. 38(3): 105–113.
Pate R.R., O’Neill J.R., and Lobelo F. 2008. The evolving definition of “sedentary”. Exerc. Sport Sci. Rev. 36(4): 173–178
Tremblay M.S., Colley R.C., Saunders T.J., Healy G.N., and Owen N. 2010. Physiological and health implications of a sedentary lifestyle. Appl. Physiol. Nutr. Metab. 35(6): 725–740.
Church T.S., Martin C.K., Thompson A.M., Earnest C.P., Mikus C.R., and Blair S.N. 2009. Changes in weight, waist circumference and compensatory responses with different doses of exercise among sedentary, overweight postmenopausal women. PLoS ONE, 4(2): e4515.
Melanson E.L., Gozansky W.S., Barry D.W., MacLean P.S., Grunwald G.K., and Hill J.O. 2009. When energy balance is maintained, exercise does not induce negative fat balance in lean sedentary, obese sedentary, or lean endurance-trained individuals. J. Appl. Physiol. 107(6): 1847–1856.
Sims S.T., Larson J.C., Lamonte M.J., Michael Y.L., Martin L.W., Johnson K.C., et al. 2012. Physical activity and body mass: changes in younger vs. older postmenopausal women. Med. Sci. Sports Exerc. 44(1): 89–97.
Smith A.E., Lockwood C.M., Moon J.R., Kendall K.L., Fukuda D.H., Tobkin S.E., et al. 2010. Physiological effects of caffeine, epigallocatechin-3-gallate, and exercise in overweight and obese women. Appl. Physiol. Nutr. Metab. 35(5): 607–616.
Tremblay, M.S., Aubert, S., Barnes, J.D. et al. Sedentary Behavior Research Network (SBRN) – Terminology Consensus Project process and outcome. Int J Behav Nutr Phys Act 14, 75 (2017). https://doi.org/10.1186/s12966-017-0525-8
Location: Line 63-69.
Q2: Over a period of time, it is observed that there is an individualised spectrum of physical activity that affects the BMI and thus overall probability of metabolic syndrome.
A2: Thank you for your practical suggestions, this is indeed an important research area that we ignored in our manuscript. We have added this in the introduction section.
Location: Line 82-85.
Round 2
Reviewer 1 Report
Thank you for addressing the concerns, which mainly focused on minor edits and clarifications. Overall, this is an important bibliometric piece on SB-CVD research.
Author Response
Dear academic editor,
Thank you for your practical suggestions. We have modified that issue in the attached document. In the modified Fig 2 , we stop at 2021 and remove data achieved in 2022. And we have mentioned: "The formula predicted that 187 papers would be published in the whole year 2022, while as of June 30th 2022, 74 documents have been published. " in our text.
Best wishes,
Zhen

Reviewer 2 Report
Article has included essential details and is reasonable to correlate findings. Author should expand the future directions of the implications of this project.